# Helicity: A Non-Conventional Stereogenic Element for Designing Inherently Chiral Ionic Liquids for Electrochemical Enantiodifferentiation

**DOI:** 10.3390/molecules26020311

**Published:** 2021-01-09

**Authors:** Francesca Fontana, Greta Carminati, Benedetta Bertolotti, Patrizia Romana Mussini, Serena Arnaboldi, Sara Grecchi, Roberto Cirilli, Laura Micheli, Simona Rizzo

**Affiliations:** 1Dipartimento di Ingegneria e Scienze Applicate, Università di Bergamo, Viale Marconi 5, 24044 Dalmine, Italy; greta.mirko@gmail.com (G.C.); benedetta.bertolotti@unibg.it (B.B.); 2CSGI Bergamo R.U., Viale Marconi 5, 24044 Dalmine, Italy; 3Dipartimento di Chimica, Università Degli Studi di Milano, Via Golgi 19, 20133 Milano, Italy; patrizia.mussini@unimi.it (P.R.M.); serena.arnaboldi@unimi.it (S.A.); sara.grecchi@unimi.it (S.G.); 4Centro Nazionale per Il Controllo e la Valutazione dei Farmaci, Istituto Superiore di Sanità, Viale Regina Elena 299, 00161 Rome, Italy; roberto.cirilli@iss.it; 5Dipartimento di Scienze e Tecnologie Chimiche, Università Degli Studi di Roma Tor Vergata, Via della Ricerca Scientifica, 1, 00133 Roma, Italy; laura.micheli@uniroma2.it; 6CNR Istituto di Scienze e Tecnologie Chimiche “Giulio Natta”, Via C. Golgi 19, 20133 Milano, Italy

**Keywords:** azahelicenes, ionic liquids, enantiodifferentiation, chiral additives, inherent chirality, chiral voltammetry

## Abstract

Configurationally stable 5-aza[6]helicene (**1**) was envisaged as a promising scaffold for non-conventional ionic liquids (IL)s. It was prepared, purified, and separated into enantiomers by preparative HPLC on a chiral stationary phase. Enantiomerically pure quaternary salts of **1** with appropriate counterions were prepared and fully characterized. *N*-octyl-5-aza[6]helicenium bis triflimidate (**2**) was tested in very small quantities as a selector in achiral IL media to perform preliminary electrochemical enantiodifferentiation experiments on the antipodes of two different chiral probes. The new organic salt exhibited outstanding enantioselection performance with respect to these probes, thus opening the way to applications in the enantioselective electroanalysis of relevant bioactive molecules.

## 1. Introduction

Some of us recently unveiled the amazing potential of chiral electroanalysis [1,2,3] (in remarkable analogy with chiroptics [4,5]) for “inherently chiral” functional compounds, in which the stereogenic scaffold responsible for chirality and the molecular group responsible for their specific properties coincide. This structural combination invariably results in outstanding enantioselection properties that are much greater than those exhibited by compounds in which the stereogenic unit and functional group are independent molecular portions. This strategy was at first implemented in terms of inherently chiral thiophene-based oligomer electrode surfaces. They were prepared using the electrochemical or chemical oxidation of suitable inherently chiral monomers. In most cases these presented with an atropisomeric biheteroaromatic system as the stereogenic element, as in the BT_2_T_4_ (Figure 1) proof-of-concept case [3], or a helix in a recent case [6].

More recently, the inherent chirality strategy was once again proven to be a winning approach when implemented in ionic liquid media. The proof-of-concept case was based on a family of 3,3′-bicollidinium salts featuring an atropisomeric biheteroaromatic system as a stereogenic element (Alk_2_**BicX_2_**, Figure 1) [1]. Impressive potential differences were observed for the enantiomers of chiral probes in voltammetry experiments on unmodified electrodes, performed in achiral ionic liquids with inherently chiral molecular salts as chiral additives. The strategy worked with different family members, including those solid at room temperature [1,2], and those with long alkyl chains and bistriflimide anions which were liquid at room temperature [1] and can be regarded as inherently chiral ionic liquids (ICILs).

Within this framework, we decided to implement the inherent chirality concept in a new family of ICILs based on the helix as an ideal stereogenic element in an inherent chirality design, and to investigate whether the great enantiodiscrimination ability of the ILs based on a stereogenic axis was also exhibited by helical organic salts.

Aza[*n*]helicenes are a class of chiral multinuclear molecules possessing peculiar electronic and chiroptical characteristics due to their extended conjugated aromatic system associated with a central distortion from planarity. They are configurationally stable above room temperature only when the number of condensed six-membered aromatic rings is ≥6 [7].

In contrast to carbohelicenes, azahelicenes possess one or more nitrogen atoms in their molecular framework which can be exploited to direct the reactivity of these molecules. Furthermore, since the nitrogen atom of azahelicenes can react with alkylating agents to afford quaternary salts, we envisaged the possibility of obtaining a completely new class of CILs. If their melting point could be lowered below 100 °C, aza-[6]helicenium cations would clearly fulfil the requirements to be defined as ICILs, since the pyridinium unit is essential part of the helical scaffold. These salts are reasonably soluble in polar solvents and exhibit physical properties that could be modulated by varying the length and shape of the alkyl moiety as well as the counterion [8]. This latter point is known to be particularly important for determining the melting range of the organic salts: the use of appropriate counteranions, like bistriflimidate, can lower the melting point by many dozens of degrees compared to halide salts [1,9].

As mentioned above, in the case of solid salts, even those with a high melting point, they could be employed as chiral dopants of achiral ILs, in which they should be perfectly soluble considering their structural affinity. We have already found that bicollidinium ICILs, when used as low-concentration additives to commercial achiral ILs, can impart high enantioselection properties towards the antipodes of chiral probes differing in structure, functional groups, and stereogenic elements [1,10]. This project could open the way towards new tools for enantioselective electroanalysis of bioactive molecules.

## 2. Results and Discussion

The molecules chosen to verify the viability of the approach were the quaternary salts of 5-aza[6]helicene (**1**), a configurationally stable azahelicene.

The synthesis of **1** was realized following the procedure reported in the literature [11]. It began with Wittig condensation between benzyl-triphenylphosphonium bromide and 4-tolualdehyde, followed by photochemical ring closure to yield 2-methylphenanthrene; the latter was brominated with N-bromosuccinimide (NBS) and the resulting bromomethyl derivative converted into the corresponding triphenylphosphonium salt. This, in turn, underwent Wittig condensation with commercially available quinoline-3-carboxaldehyde to yield 1-(2-phenanthrenyl)-2-(3-quinolyl)ethane **3**, mainly as *E* isomer, in 71% yield (Scheme 1).

The final step was again a photochemical ring closure, which, however, unexpectedly led to two constitutional isomers, namely **1** and **4** (Scheme 2).

In fact, precursor **3**, which was formed as an *E/Z* mixture, during photolysis isomerized to the *Z* stereoisomer, for which two conformational *s*-cis (**3a**) and *s*-trans (**3b**) isomers were expected. Oxidative photocyclization of the former afforded 5-aza[6]helicene (**1**), while ring closure of the latter led to the formation of the achiral phenanthreno[2,3-k]phenanthridine (**4**) (Scheme 2).

When performing the analytical HPLC separation of the racemate of **1** on chiral stationary phase (see below), we observed the presence of 39% of **4**, previously undetected due to its very high gas-chromatographic retention time. At this point, while proceeding with the experiments with the isolated enantiomers, we undertook the task of improving the synthesis of compound **1** by reducing or eliminating the formation of byproduct **4**. We found rather difficult to separate **4** from racemic **1** by column chromatography on silica gel and minimize its formation by varying the reaction conditions. We attempted to perform photochemical ring closure of **3** in different solvents as previous investigations on a diazapentahelicene had evidenced how this step could be preferentially directed towards the formation of the azahelicene by the use of polar solvents [12]. However, in this case, the use of acetonitrile instead of ethyl acetate resulted in the almost exclusive formation of **4**, while using dichloromethane or toluene we obtained both products with a prevalence of **4**. We eventually managed to obtain an almost complete prevalence of **1** by using a 9:1 hexane/ethyl acetate mixture, where ethyl acetate was at just the minimum quantity necessary for the solubilization of **3**. However, the yield of **1** in these conditions was 34%, with just traces of **4**, while the remaining starting material underwent degradation during the process.

In order to avoid this drawback, we explored the possibility of approaching the synthesis of **1** through the alternative pathway depicted in Scheme 3 based on the Heck condensation. 11-[(*E*)-styryl]benzo[k]phenanthridine **5** was obtained as an *E/Z* isomeric mixture and its photocyclization exclusively provided the desired product **1**. It is to be noted that in this case two conformational isomers would also be possible, one of which would bring the formation of **4**.

Product **5** was also prepared by using the classical Wittig approach between the four-rings phosphonium ylide moiety and benzaldehyde, but overall yields were considerably lower than with the Heck approach. The yield of the photocyclization of **5** at 366 nm to product **1** was 90%.

The racemate of **1** obtained by photolysis of intermediate **3** in ethyl acetate solution was resolved into enantiomers by semipreparative HPLC on Chiralpak IA chiral stationary phase; the process also allowed the total removal of the achiral product **4** from the enantiopure antipodes of **1** (Figure 2) and its isolation in a high-purity state.

In accordance with the circular dicroism (CD) data in the literature, [11], the absolute *P* configuration was assigned to the first eluted enantiomer.

Both enantiomers were alkylated with octyl iodide excess at 80 °C, followed by anion metathesis with silver bis(trifluoromethanesulfonyl)amide AgNTf_2_ to provide enantiomerically pure quaternary salts as waxy solids (Scheme 4). The starting material configuration is supposed to be retained in the corresponding salts on account of the high racemization barrier of **1** (the enantiomeric excess of **1** remains unchanged after 24 h heating in dimethylformamide solution at 100 °C).

Then, **1** and **2** were electrochemically characterized by cyclic voltammetry (CV, Figure 3). As a first general consideration, in the parent azahelicene the nitrogen lone pair in the pyridine ring provides a preferential site for first oxidation, resulting in radical cation formation. However, upon alkylation the situation is reversed, since the nitrogen becomes cationic and hence much more electron-poor, thus providing preferential sites of first reduction (similarly e.g., to the effect of alkylation on the electrochemical behavior of other pyridine and benzimidazole scaffolds [1,13]). Cation reduction leads to radical formation, with possible coupling follow-ups, resulting e.g., in dimerization, as reported for the 1-methyl-1-aza[6]helicenium cation [14].

Comparing the CV features of **1** and **2**, the alkylation of the aza site resulted in:(1)The disappearance of the first irreversible oxidation peak at 1.07 V;(2)The disappearance of most of the complex reduction peak system between −2.2 and −2.7 V, leaving a single nearly chemically and electrochemically reversible peak at −2.58 V;(3)The appearance of a nearly irreversible and nearly splitting first reduction peak at −1.19 V.

Both the reduction peak systems of **2** were at potentials very close to those reported for the *N*-methyl-1-aza[6]helicenium cation [14]. Furthermore, a peak, surely associated with reoxidation of reduced products on the electrode surface, could be seen for **2** at about −0.45 V in a scan starting from reduction and also including oxidations (Figure 4a) (a second oxidative return peak at about 0 V was instead associated with the only partially reversible second reduction peak, Figure 4b). Thus, reversible dimer formation as in [14], at least as partial follow-up of the first reduction process, cannot be ruled out. An additional similarity with a case in the literature [14] is that the first reduction peak of **2** tended to split or to feature a preceding shoulder, as confirmed in two experiments at different times (see Appendix A). Notably, the same feature also appeared in the CV patterns of the azahelicenium compound described in [14], although the authors do not specifically mention it. Since we cannot assume any conformational isomerism for the single molecule, the above features might be linked to two pathways with different preceding or following chemical steps (for example, partial coupling as in [10]), besides solid state adsorption or aggregation effects.

The irreversible oxidation peaks at 1.46 V in **1** (the second one after aza oxidation) and 1.47 V in **2** (only one) could instead be ascribed to the carbahelicene terminal, considering its consistency with the reported first oxidation potential of 1.405 V for 6-carbahelicene [15]. The small positive shift could be justified for **1** considering that in that case the electron transfer process takes place after a former one, that is, with a positive charge already present on the molecule, and for **2** since the molecule is electron-poorer on account of the alkylated aza site.

The nearly reversible (both chemically and electrochemically) second reduction peak of **2** at −2.6 V is more likely to be related to first reduction (to radical anion) of the carbahelicene terminal than to a second reduction on the aza site, since in 6-carbahelicene the first reduction is at an even more positive potential [15]. Consistently, the superimposed peak system appearing in the case of **1** should correspond to the reduction of the opposite helicene side, the one with the pyridyl ring adjacent to the phenyl terminal.

To check the enantioselection ability of the (*P*)-**2** and (*M*)-**2** salts, we first tested them as 0.01 M additives in achiral ionic liquid (BMIM)NTf_2_, recording in small volumes of the resulting chiral media the CV patterns of the (*R*)-(+)- and (*S*)-(−)-*N*,*N*′-dimethyl-1-ferrocenylethylamine chiral probe enantiomers on screen-printed electrode (SPE) cells with an Au working electrode (Figure 5).

Despite the medium viscosity, neat CV peaks (chemically and electrochemically reversible or quasi-reversible) were obtained for all probe+selector combinations (Figure 5). At the same time, a remarkable difference was observed for the enantiomer peak potentials (~140 mV) together with slight differences in the peak shape (for example, there was a greater difference between forward and backward peak potentials for the peak at higher potential). In particular, the oxidation peak potential of one enantiomer remained approximately in the same position, while the other shifted to much higher potentials, while maintaining its reversible peak shape, a feature often linked to preferential stabilization of the reactant (or destabilization of the product). Neatly specular features were obtained inverting either selector or probe configuration, while the (*R*)- and (*S*)-probe enantiomers gave practically coincident CV peaks when performing the same protocol in achiral (BMIM)NTf_2_ in the absence of the chiral additive. Repetitions on new SPE supports were performed in all cases in order to check the result repeatability (Figure 5).

A second series of enantioselection experiments was performed using differential pulse voltammetry (DPV) on SPE cells with working graphite electrodes using the (*P*)-**2** salt as 0.01 M additive in achiral (BMIM)NTf_2_ to test the (*R*)- and (*S*)-enantiomers of tyrosine methyl ester hydrochloride at 0.002 M concentration. Once more, a neat and fairly reproducible peak potential difference of ~130 mV was observed for the first oxidation peaks in the CV patterns of each enantiopure probe, with a constant working protocol, while the same probe enantiomers gave practically the same peak in the absence of the chiral additive in the achiral IL medium (Figure 6).

It is interesting to compare the effects produced on enantioselectivity by ICILs characterized by different stereogenic elements, namely a stereogenic axis and a helix, when used as chirality inducers in low concentration in achiral ILs. The terms of comparison were the *N*-monoalkyl- and the *N**,N’*-dialkyl-3,3′-bicollidinium salts (Figure 1), C_2_ symmetric when the alkyl chains are identical and asymmetric when the alkyl groups are different. Under comparable experimental conditions and using the same ferrocenylamine probes, the bicollidinium salt represented in Figure 1, characterized by two identical *n*-octyl groups, at a 0.01 M concentration developed a potential difference higher than 170 mV [1], while the asymmetric bicollidinium compound with the nitrogen atoms differently quaternarized with a methyl and a *n*-octyl group respectively afforded a 120-mV peak potential separation at a 0.016 M concentration. Therefore, in this first attempt to verify the enantioselection obtainable by azahelicenium salts, we observed a potential difference comparable to that attainable with other inherently chiral additives, such as bicollidinium compounds.

## 3. Materials and Methods

### 3.1. General Procedures

The starting materials and solvents for azahelicene synthesis were purchased from Carlo Erba (Milan, Italy) and used without further purification; photolysis reactions were performed using a Multirays instrument equipped with sets of 10 lamps of different wavelengths. NMR spectra were recorded on Bruker AV400 and Bruker AC300 spectrometers. Chemical shifts (δ) are expressed in parts per million (ppm), and coupling constants are given in Hz. Splitting patterns are indicated as follows: s = singlet, d = doublet, t = triplet, q = quartet, quint = quintet, m = multiplet. Purifications by column chromatography were performed using Merck silica gel 60 (230–400 mesh for flash-chromatography and 70–230 mesh for gravimetric chromatography) and aluminium oxide 90 neutral. Melting points were determined on a Büchi B-540 instrument. GC-MS analyses were performed on an Agilent 6850 chromatograph equipped with an Agilent 5975N mass spectrometer.

HPLC-grade solvents were purchased from Sigma-Aldrich (Milan, Italy). HPLC enantio-separations were performed by using stainless-steel Chiralpak IA (250 mm × 4.6 mm, 5 μm and 250 mm × 10 mm, 5 μm) columns (Chiral Technologies Europe, Illkirch, France). The HPLC apparatus used for analytical enantioseparations consisted of a PerkinElmer (Norwalk, CT, USA) 200 LC pump equipped with a Rheodyne (Cotati, CA, USA) injector, a 50-μL sample loop, an HPLC PerkinElmer oven, and a Jasco (Jasco, Tokyo, Japan) Model CD2095 Plus UV/CD detector. The signal was acquired and processed by Clarity software (DataApex, Prague, Czech Republic). For semipreparative separation, a PerkinElmer 200 LC pump equipped with a Rheodyne injector, a 5000-μL sample loop, a PerkinElmer LC 101 oven, and a Waters 484 detector (Waters Corporation, Milford, MA, USA) were used.

### 3.2. Electrochemistry

#### 3.2.1. Helicene Characterization

Cyclic voltammetry (CV) was performed at scan rates in the range of 0.05–2 V/s using an AutoLab PGStat potentiostat and a classical three-electrode glass minicell (with a working volume of about 3 cm^3^). The latter included as a working electrode a glassy carbon (GC) disk embedded in glass (Metrohm) polished by diamond powder (1 μm Aldrich) on a wet cloth (Struers DP-NAP), as counter electrode a platinum disk, and as a reference electrode a saturated aqueous calomel electrode (SCE) inserted into a compartment with the working medium ending in a porous frit to avoid contamination of the working solution by water and KCl traces. Experiments were run with 0.00075 M azahelicene solutions in acetonitrile (ACN, Aldrich, HPLC grade) + 0.1 M tetrabutylammonium hexafluorophosphate TBAPF_6_ (Fluka, ≥98 %) as the supporting electrolyte, previously deaerated by nitrogen bubbling. (*S*)-L-Tyr Me Ester was purchased from Sigma-Aldrich and (*R*)-D-Tyr Me Ester from Alfa Aesar.

#### 3.2.2. Enantiodiscrimination Experiments

CV enantiodiscrimination tests were performed by cyclic voltammetry at a 0.05 V/s scan rate on screen-printed electrode (SPE) supports (Dropsens, custom made without paint, with Au working and counter electrodes and an Ag pseudoreference electrode, resulting in good reproducibility at constant conditions with the present working protocol).

The experiments were performed using (*P*)-**2** or (*M*)-**2** salts as 0.01 M chiral additives in achiral commercial IL 1-butyl-3-methylimidazolium bis(trifluoromethanesulfonyl)imidate (BMIM)NTf_2_ (CAS 174899-83-3; Aldrich 98%) with the same counteranions.

CVs were recorded in open air conditions, depositing on the working electrode a drop of one of the above chiral media with 0.002 M (*R*)- or (*S*)-antipodes of *N,N*′- dimethyl-1-ferrocenylethylamine (Aldrich, submitted to a further chromatographic purification step), usually employed as model chiral probe by some of us when testing “inherently chiral” electrode surfaces and media on account of its chemical and electrochemical reversibility.

A second series of enantioselection experiments was performed using differential pulse voltammetry (DPV) on laboratory screen-printed electrode cells on a plastic (polyester) sheet with an insulating layer including graphite working and counter electrodes and an Ag pseudoreference, resulting in good reproducibility at constant conditions with the present working protocol. Again, the experiments were performed in open air conditions, using the (*P*)-**2** salt as the 0.01 M chiral additive in achiral commercial IL (BMIM)NTf_2_ to test (*R*)- and (S)-enantiomers of tyrosine methyl ester hydrochloride ((*S*)-L-Tyr Me Ester from Aldrich; (*R*)-D-Tyr Me Ester from Alfa Aesar) at 0.002 M concentration.

### 3.3. Synthesis of 5-aza[6]helicene

The synthesis was realized as described in [11]. Precursor **3**, which is formed as a mixture of *Z* and *E* isomers, was photolysed in various solvents or solvent mixtures in Pyrex vessels with visible light for 24 h. When the photolysis took place in a hexane - ethyl acetate 9:1 a mixture of the desired product (**1**) was obtained in 34% yield, with traces of its isomer (**4**) (Scheme 2), The byproduct proved difficult to separate completely from the azahelicene by column chromatography; after many attempts, separation was obtained by using alumina as stationary phase and by eluting with hexane/ethyl ether 4:1.

*(±) phenanthro[3,2-k]phenanthridine* (**4**). ^1^H NMR (300.14 MHz, HZ/pPT(Hz) = 0.15, CDCl_3_) δ 9.69 (s, 1H), 9.39 (s, 1H), 9.30 (s, 1H), 9.28 (d, ^3^*J* = 2.7 Hz, 1H), 8.91 (d, ^3^*J* = 8.1 Hz, 1H), 8.44–8.36 (m, 1H), 8.24 (d, ^3^*J* = 8.7 Hz, 1H), 8.07–7.92 (m, 3H), 7.91–7.66 (m, 5H); ^13^C NMR (75.48 MHz, HZ/pPT(Hz) = 0.13, CDCl_3_) δ 151.94 (s), 146.43 (s), 133.30 (s), 132.42 (s), 131.69 (s), 131.13 (s), 130.10 (s), 129.95 (s), 129.48 (s), 128.77 (s), 128.48 (s), 128.35 (s), 127.97 (s), 127.76 (s), 127.45 (s), 127.31 (s), 127.14 (s), 126.77 (s), 125.05 (s),124.82 (s), 124.75 (s), 123.25 (s), 122.58 (s).

The alternative synthesis of the same product (Scheme 3) was realized by refluxing p-bromobenzylphosphonium bromide (3.9 mmol) and 3-quinolinecarboxaldehyde (3.9 mmol) with 11.7 mmol (3 eq) of t-BuOK in 50 mL of methanol overnight. Methanol was evaporated and the crude dissolved in ethyl acetate and washed with water. The organic phase was dried on anhydrous Na_2_SO_4_, evaporated and purified by column chromatography on silica gel, eluting with hexane/ethyl acetate 1:1. The product, p-bromophenyl-3-quinolyl-ethane, was obtained in 79% yield. It was then photolyzed in ethyl acetate in quartz vessels with 366-nm lamps for 2.5 h, in the presence of catalytic I_2-_. The photolyzed solutions were united, evaporated and chromatographed on silica gel with hexane/ethyl acetate 1:1. The yield of 11-bromobenzo[k]phenanthridine was 86%. This latter product was then reacted with styrene (4 eq) in dimethylacetamide (DMA, 22 mL) in the presence of 0.01 eq of (Ph_3_P)_2_PdCl_2_ and 3 eq of sodium acetate trihydrate, under nitrogen atmosphere at 140 °C for 2 days. The solution was cooled at room temperature, diluted with AcOEt, washed with water (three times) to eliminate the DMA, evaporated, and chromatographed on silica gel with hexane/ethyl acetate 1:1. The yield of product **5** was 94%.

Product **5** was then dissolved in ethyl acetate (0.5 mg/mL) and photolyzed at 366 nm for 2 h. The solution was then evaporated and product **1** purified by column chromatography over silica gel, eluting with hexane-acetate 1:1. Yield 90%

### 3.4. Synthesis of (P)-5-octyl-5-aza[6]helicenium Iodide

The iodooctane (0.4 mL) was added to (*P*)-5-aza[6]helicene (3.31 mg, 0.01 mmol). The reaction mixture was heated at 80 °C for 56 h, then the crude was washed with hexane. The product was obtained (5.3 mg, 93%). ^1^H NMR (300.14 MHz, HZ/pPT(Hz) = 0.09, CDCl_3_) δ 11.41 (s, 1H), 8.88 (d, ^3^J = 8.1 Hz, 1H), 8.39 (d, ^3^J = 8.1 Hz, 1H), 8.32 (d, ^3^J = 8.1 Hz, 1H), 8.18 (d, ^3^J = 9.0 Hz, 1H), 8.15 (d, ^3^J = 8.4 Hz, 1H), 8.10 (d, ^3^J = 8.7 Hz, 1H), 8.03 (d, ^3^J = 9.0 Hz, 1H), 7.95 (d, ^3^J = 7.8 Hz, 1H), 7.79 (d, ^3^J = 8.4 Hz, 1H), 7.70 (t, ^3^J = 7.8 Hz, 1H), 7.48 (d, ^3^J = 8.4 Hz, 1H), 7.37 (t, ^3^J = 7.5 Hz, 1H), 7.04 (t, ^3^J = 7.8 Hz, 1H), 6.87 (t, ^3^J = 7.7 Hz, 1H), 5.47 (t, ^3^J = 7.5 Hz, 2H), 2.33 (quint, ^3^J = 7.5 Hz, 2H), 1.80-1.19 (m, 10H), 0.89 (t, ^3^J = 6.6 Hz, 3H); APT NMR (75.48 MHz, HZ/pPT(Hz) = 0.14, CDCl_3_) δ 152.82 (s), 138.55 (s), 135.38 (s), 133.67 (s), 132.78 (s), 132.42 (s), 132.34 (s), 131.35 (s), 131.00 (s), 129.90 (s), 128.94 (s), 128.40 (s), 127.84 (s), 127.76 (s), 127.54 (s), 127.09 (s), 126.95 (s), 126.12 (s), 125.82 (s), 125.40 (s), 124.97 (s), 123,25 (s), 121,49 (s), 117.69 (s), 57.51 (s), 31.64 (s), 31.53 (s), 30.56 (s), 29.14 (s), 29.05 (s), 26.59 (s), 22.52 (s), 14.00 (s).

### 3.5. Synthesis of (P)-5-octyl-5-aza[6]helicenium bis(trifluoromethanesulfonyl)imidate

A solution of silver bis(trifluoromethanesulfonyl)imide (3.06 mg, 7.9 × 10^−3^ mmol) in EtOH (0.2 mL) was dropped into a stirred solution of (*P*)-5-octyl-5-aza[6]helicenium iodide (4.5 mg, 7.9 × 10^−3^ mmol) in EtOH (0.2 mL). Stirring was continued for 72 h at room temperature to complete the reaction. During this time the precipitate was formed, and this was removed by filtration. Finally, the filtrate was evaporated under reduced pressure and the corresponding product was obtained as a waxy solid (4.8 mg, 84%). ^1^H-NMR (300.14 MHz, HZpPT(Hz)=0.15, CDCl_3_) δ 10.14 (s, 1H), 8.55 (d, ^3^*J* = 8.4 Hz, 1H), 8.39 (d, ^3^*J* = 8.4 Hz, 1H), 8.33 (d, ^3^*J* = 8.4 Hz, 1H), 8.17 (d, ^3^*J* = 8.4 Hz, 1H), 8.15 (d, ^3^*J* = 8.1 Hz, 1H), 8.11 (d, ^3^*J* = 8.7 Hz, 1H), 8.03 (d, ^3^*J* = 8.7 Hz, 1H), 7.95 (d, ^3^*J* = 7.8 Hz, 1H), 7.80 (d, ^3^*J* = 8.7 Hz, 1H), 7.72 (t, ^3^*J* = 7.4 Hz, 1H), 7.44 (d, ^3^*J* = 8.7 Hz, 1H), 7.37 (d, ^3^*J* = 7.5 Hz, 1H), 7.04 (d, ^3^*J* = 7.7 Hz, 1H), 6.85 (d, ^3^*J* = 7.8 Hz, 1H), 5.32-5.12 (m, 2H), 2.27 (quint, ^3^*J* = 7.6 Hz, 2H), 1.70-1.52 (m, 2H),1.52-1.20 (m, 8H), 0.89 (t, ^2^*J* = 6.6 Hz, 3H); ^19^F NMR (300.14 MHz, HZ/pPT(Hz) = 0.87, CDCl_3_) δ −78.0 (s); APT NMR (100.62 MHz, HZ/PT(Hz)=1.47, CDCl_3_); δ 152.46 (s), 138.82 (s), 135.68 (s), 133.55 (s), 132.84 (s), 132.62 (s), 132.47 (s), 131.54 (s), 131.47 (s), 130.11 (s), 129.15 (s), 128.51 (s), 128.07 (s), 127.90 (s), 127.54 (s), 127.50 (s), 127.12 (s), 126.14 (s), 125.86 (s), 125.58 (s), 123.22 (s), 121.60 (s), 119.82 (q, ^2^*J*(C,F) = 322.0 Hz), 117.61 (s), 58.26 (s), 31.62 (s), 30.20 (s), 29.03 (s), 26.53 (s), 22.55 (s), 14.02 (s). APT NMR Only Quaternary Carbon (100.62 MHz, HZ/PT(Hz)=1.47, CDCl_3_); δ 138.84 (s), 135.68 (s), 132.86 (s), 132.62 (s), 132.49 (s), 128.54 (s), 127.92 (s), 125.60 (s), 123.27 (s), 121.62 (s), 119.87 (q, ^2^*J*(C,F) = 321.0 Hz).

### 3.6. Enantioselective HPLC

Column: Chiralpak IA 250 mm × 4.6 mm, mobile phase: *n*-hexane-IPA-ethyl acetate-DEA 100/5/5/0.2, flow rate: 1 mL/min, detector: UV/CD at 325 nm.

The CD spectra of the enantiomers collected on a semipreparative scale were recorded in chloroform at 25 °C by using a Jasco Model J-700 spectropolarimeter. The optical path was 0.1 cm. The spectra are average computed over three instrumental scans and the intensities are presented in terms of ellipticity values (mdeg).

### 3.7. Off-Column Racemization Study

A solution of (*P*)-**1** in dimethylformamide (concentration about 0.2 mg/mL) was held at 100 °C in a closed vessel. The temperature was monitored by a thermostat Julabo HE-4. Samples were withdrawn at fixed time intervals and the ee decay over time was monitored by HPLC on the Chiralpak IA (250 mm × 4.6 mm i.d.) column under normal-phase mode.

## 4. Conclusions

A new ICIL was synthesized, characterized, and successfully tested as additive of a commercial IL for electrochemical enantiodiscrimination purposes. In fact, significant peak potential differences were reproducibly observed in CV or DPV experiments for the enantiomers of two different chiral electroactive probes.

The conclusions that we can draw from this very preliminary experiment are twofold.

The first consideration is that inherent chirality extraordinarily enhances the enantioselection ability in comparison with chiral ILs designed according to more traditional schemes involving the separation between chiral and onium moieties [16]. In fact, peak potential separations jump here from a few dozen mV to more than 100.

The second point is that helicity induces enantioselectivities comparable to those produced by stereogenic axes. The helix can be regarded as an ideal stereogenic element in the design of inherently chiral selectors, even though the access to enantiopure helical systems is definitely more troublesome than the synthesis of chiral compounds endowed with axial stereogenicity.

We plan to extend the study to diazahelicenes quaternarized on one or both nitrogen atoms in order to compare them to the mono- and dialkylated bicollidinium compound.

## Data Availability

Data sharing not applicable.

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
