# Peer review of "Helicity: A Non-Conventional Stereogenic Element for Designing Inherently Chiral Ionic Liquids for Electrochemical Enantiodifferentiation"

_molecules, 2021, doi:10.3390/molecules26020311_

Round 1
Reviewer 1 Report
The authors presented a well-documented study on using inherently chiral N-alkylated aza-helicene as an additive in CV run in the liquid-crystalline system. The reported results are convincing and can be considered as proof of concept. The paper is well-written and easy to follow.
However, more experiments are needed to develop an analytical methodology, allowing for determining the configuration and ee of the solute (analyte).
Author Response
the Authors appreciate the favorable comments of the Referee. As for the comment: “more experiments are needed to develop an analytical methodology, allowing for determining the configuration and ee of the solute (analyte)” the paper was actually meant as a proof-of-concept, as observed by the Referee. Indeed, the enantioselectivity experiments proved reproducible and showed a significant potential shift for two different enantiomeric probes. It is our intention to further explore the potentiality of the method and, in future works, to optimize the analysis protocol and verify the possibility, already observed in other instances, to determine the ee of the analyte, particularly in the case of a reversible CV experiment.
Reviewer 2 Report
I find the paper very nice nad well written, reporting important results. I just hve some minor comments.
Concerning the compounds purity, it would be nice to have the NMR spectra (with peak identification) in the Supporting Information (the standard template text was left in that section).
Also it would be useful if elemental analyses results were added to the paper.
There are some parts in green highlight, I believe authors forgot to remove.
Author Response
We checked the manuscript for the green highlights (lines 302 and 383) which were not removed prior to submission. We also completed the Supporting Information section with the numbers and captions of the Figures (lines 392-400) We thank the referee for pointing it out. As for the NMR and elemental analysis, we can certainly provide them; however, since in these days our University is closed for vacations, we will not be able to provide them within a few days; we might add the required items to the SI as soon as we can have access to the labs.